# Virtual Visual-Guided Domain-Shadow Fusion via Modal Exchanging for Domain-Specific Multi-Modal Neural Machine Translation

Zhenyu Hou
Faculty of Information Engineering and Automation,
Kunming University of Science and Technology
Kunming, Yunnan, China
hzy23@stu.kust.edu.cn

Junjun Guo*
Faculty of Information Engineering and Automation,
Kunming University of Science and Technology
Kunming, Yunnan, China
guojjgb@163.com

## ABSTRACT

Incorporating domain-specific visual information into text poses one of the critical challenges for domain-specific multi-modal neural machine translation (DMNMT). While most existing DMNMT methods often borrow multi-modal fusion frameworks from multimodal neural machine translation (MNMT) in the general domain, they overlook the domain gaps between general and specific domains. Visual-to-textual interaction in a specific domain frequently exhibits multi-focus characteristics, making it difficult to consistently focus on domain-specific multi-visual details using traditional multi-modal fusion frameworks. This challenge can lead to a decrease in machine translation performance for domain-specific terms. To tackle this problem, this paper presents a virtual visual scene-guided domain-shadow multi-modal fusion mechanism to simultaneously integrate multi-grained domain visual details and text with the guidance of modality-agnostic virtual visual scene, thereby enhancing machine translation performance for DMNMT, especially for domain terms. Specifically, we first adopt a modality-mixing selection-voting strategy to generate modality-mixed domain-shadow representations through layer-by-layer intra-modality selection and inter-modality exchanging. Then, we gradually aggregate modality-mixed domain representations and text across modality boundaries with the guidance of a modality-agnostic virtual visual scene to enhance the collaboration between domain characteristics and textual semantics. The experimental results on three benchmark datasets demonstrate that our proposed approach outperforms the state-of-the-art (SOTA) methods in all machine translation tasks. The in-depth analysis further highlights the robustness and generalizability of our approach across various scenarios. Our code is available on https://github.com/HZY2023/VVDF.

## CCS CONCEPTS

• **Computing methodologies** → **Machine translation**; *Computer vision representations*; Natural language generation.

*corresponding author

## KEYWORDS

Domain-specific Multimodal Neural Machine Translation, Multimodal Fusion, Modality Exchanging

**ACM Reference Format:**
Zhenyu Hou and Junjun Guo. 2024. Virtual Visual-Guided Domain-Shadow Fusion via Modal Exchanging for Domain-Specific Multi-Modal Neural Machine Translation. In *Proceedings of the 32nd ACM International Conference on Multimedia (MM '24), October 28-November 1, 2024, Melbourne, VIC, Australia.* ACM, New York, NY, USA, 10 pages. https://doi.org/10.1145/3664647.3681525

## 1 INTRODUCTION

Domain-specific Multimodal Neural Machine Translation (DMNMT) aims to translate sentences within specific domains from a source to a target language by incorporating images as additional modality inputs. Recently, DMNMT tasks have attracted increasing attention, particularly due to their remarkable application scenarios in domains like cross-border e-commerce shopping, cross-border financial transactions, and cross-border cultural exchange.

Due to the similarity to multimodal neural machine translation (MNMT) [10, 42, 44], previous DMNMT works typically follow MNMT frameworks in general domain and concentrate on integrating visual and textual modalities using various cross-modal fusion strategies, such as cross-modal gating [15, 22, 42], cross-modal attention [12, 36, 45], and adaptive feature selection [9, 43]. Despite achieving impressive performance, there are significant domain gaps in general and specific domains.

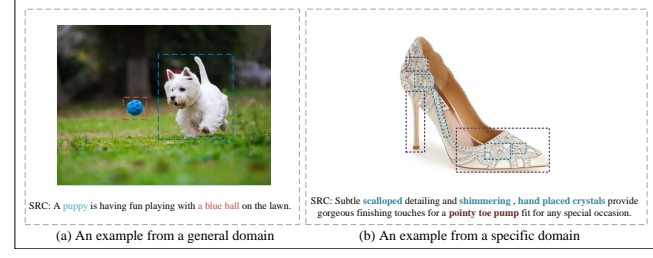

SRC: A puppy is having fun playing with a blue ball on the lawn.

SRC: Subtle scalloped detailing and shimmering, hand placed crystals provide gorgeous finishing touches for a pointy toe pump fit for any special occasion.

(a) An example from a general domain    (b) An example from a specific domain

**Figure 1: Visual-to-textual interaction of two multi-modal neural machine translation examples.**

Domain-specific sentences contain a series of domain-related expressions, such as domain entities, domain terms, and domain

idioms, which are significantly different from terms in general domain. Representation and translation of these domain-specific expressions pose critical challenges for DMNMT. Numerous studies have shown that visual information is often essential for translating domain-specific terms [33, 46], whereas it is usually optional for general-domain text [39]. Therefore, integrating domain-specific visual details into text represents an effective strategy to address this challenge.

Unfortunately, domain-related visual information tends to be fine-grained and dispersed. Figure 1 illustrates two examples of multi-modal machine translation in the general and specific domains. In contrast to multi-modal information in the general domain, the domain-specific image contains more focal areas related to domain terms in the text, such as the colored dashed regions illustrated in Figure 1 (b). Therefore, visual-to-textual interaction in a specific domain often exhibits multi-focus characteristics.

Several attempts have been made to explore domain-specific multi-modal fusion problems in DMNMT through techniques such as domain multi-modal data augmentation [46], domain-specific pretraining [33] and domain-specific multimodal feature interaction [14]. However, most existing multi-modal fusion approaches primarily follow the steps of traditional multi-modal fusion in general domain, making it challenging to consistently focus on domain-specific multi-visual details. This limitation may impact their adaptability and effectiveness for multi-modal domain representations.

Modality exchanging [37, 38] provides a potential cross-modal selection mechanism within and across modalities to aggregate these dispersed domain visual details simultaneously from domain image-text data pairs. The domain-specific multi-modal information can be aggregated from visual to textual direction through intra-modality selection and inter-modality exchanging. These modality-mixed domain details can then serve as multi-modal domain-shadow representations to augment textual domain representations, thereby enhancing domain translation performance.

However, these modality-mixed domain-shadow representations are generated through cross-modal feature direct exchanging, which may lack smoothness in multi-modal space. Therefore, there is a significant representation gap between the representations of modality-mixed domain-shadow details and raw text. The modality-agnostic virtual visual scene [25], created through visual distillation [25, 30] with textual inputs, serves as multi-modal domain guidance to enhance the generation of smoother multimodal representations for domain image-text data pairs. Inspired by this, this paper presents a virtual visual scene-guided progressive domain-shadow multi-modal fusion mechanism to gradually facilitate the integration of textual semantics with domain-specific visual details through layer-by-layer cross-modal exchanging, thereby enhancing the collaboration between domain characteristics and textual semantics. Compared to existing works, the major contributions of our paper are three-fold.

- To tackle the multi-focus challenges of visual-to-textual interaction in DMNMT, we present a novel virtual visual scene-guided progressive domain-shadow fusion mechanism aimed at integrating textual semantics with domain-specific visual details progressively to improve the model's ability to perceive fine-grained domain visual details in DMNMT. Our

proposed approach is designed to concurrently capture dispersed domain visual details through visual-textual modality mixing, and gradually aggregate modality-mixed domain representations and text with the shadow guidance of a modality-agnostic virtual visual scene. Additionally, the virtual visual scene benefits from cross-modal adaptive distillation.
- We employ a modality-mixing selection-voting strategy to aggregate domain-specific multi-modal representations through intra-modality selection and inter-modality exchanging. Initially, we introduce a fine-grained domain voting strategy to select domain visual details and domain loosely-related textual tokens in their respective modality-specific spaces. Subsequently, we facilitate the exchange of selected information across modality boundaries to generate modality-mixed domain representations by incorporating visual details into text.
- The extensive experiments on three datasets demonstrate that our proposed approach achieves state-of-the-art (SOTA) scores on two domain-specific and one in-general machine translation tasks. The in-depth analysis showcases that our proposed approach also achieves significant robustness and generalizability across various scenarios, such as noisy image-text or even text-only scenarios. Moreover, our proposed virtual visual scene generation module still exhibits strong model compression capability, underscoring its potential for practical applications.

## 2 RELATED WORK

**Multimodal Neural Machine Translation.** Recently, MNMT has drawn much attention in the field of Natural Language Processing (NLP). Existing MNMT works mainly focus on how to better integrate visual information into text to enhance the performance of machine translation. There are two types of visual information integration strategies for MNMT, including 1) image-must method: traditional MNMT works [16, 21, 36] often preferred to employ the aligned images to enhance machine translation performance through image-must multi-modal fusion strategies. Yao et al. [40] proposed a Transformer-based multimodal self-attention mechanism to address the noisy-robust multi-modal fusion problem in MNMT. Similarly, Ye et al. [41] developed a mask-guided cross-attention framework to tackle the issue of visual-textual semantic alignment in MNMT. Furthermore, Li et al. [24] devised a semi-supervised multimodal attention to fuse textual and visual modalities through cross-modal alignment. 2) image-free MNMT methods: recently, image-free MNMT methods [9, 25, 27, 30] have achieved wide attention to enhance machine translation performance through knowledge distillation. These approaches aimed to alleviate the constraints imposed by triplet data for MNMT. Long et al. [27] proposed a visual imagination method to synthesize continuous image features for machine translation. Li et al. [30] utilized a visual-hallucination method to generate discrete visual representations of text, enhancing machine translation performance. Guo et al. [16] aimed to enhance machine translation performance with the aid of synthetic representations by minimizing the semantic gaps between ground-truth and synthetic images. However, the aforementioned studies primarily focused on in-general

cross-modal fusion, ignoring the specific challenges posed by DM-NMT. How to leverage visual information to improve domain-specific machine translation performance is still an open problem.

**Knowledge Distillation.** Knowledge distillation (KD) [3, 17, 18, 30] aims to transfer knowledge from a teacher to student models, which has achieved wide attention in the fields of both computer vision and NLP. This concept was initially introduced by [3] and subsequently improved by [20]. Subsequent works have further improved the logits-based KD through structural information [13], model ensemble [28] and adversarial learning [26]. Then some approaches [1, 4] explored feature-based knowledge distillation by utilizing the intermediate representations as hint knowledge. Furthermore, cross-modal knowledge distillation [11, 23, 31] has garnered significant attention recently. Sarkar et al. [31] devised a self-supervised framework to perform effective information sharing between audio and video streams to obtain more generalized representations through cross-modal KD. IKD-MMT [30] introduced an inverse knowledge distillation framework to generate multi-modal representations according to source text. In addition, some works have applied KD to Large Language Models. For example, Li et al. [23] proposed a CLIP-based knowledge distillation hashing approach to capture the semantic relevance and coexistent information for multimodal data. However, most KD approaches often overlook the fact that textual-visual KD approaches contain valuable multimodal information during the distillation process, which could effectively guide the fusion of multimodal information.

## 3 METHODOLOGY

One of the critical challenges of DMNMT is to incorporate domain visual details into text to enhance machine translation performance. However, in contrast to multi-modal fusion in the general domain, domain-specific visual-to-textual fusion suffers from multi-focus challenges. To tackle this challenge, this paper presents a virtual visual scene-guided progressive domain-shadow multi-modal fusion approach to capture domain-specific multi-modal representations, thereby enhancing machine translation performance. The overall framework of our paper is illustrated in Figure 2, which consists of the following three subsections: 1) Modality-specific embeddings; 2) Domain-shadow aggregation with modality-mixing selection-voting strategy; and 3) Virtual visual scene-guided progressive domain-shadow fusion.

### 3.1 Modality-specific Embeddings

Denote by $\{x_k, v_k, y_k\}$ as the $k$-th domain data pair, where $x_k = \{x_1, \ldots, x_n^k\}$ and $y_k = \{y_1, \ldots, y_m^k\}$ denote the domain-specific source and target sentences, respectively. $v_k$ represents their corresponding image, $n$ and $m$ are the lengths of $x_k$ and $y_k$.

*3.1.1 Textual Embedding.* We first leverage the textual embedding module to extract initial textual representation, as shown as follows:

$$E_x = \text{Emb}_s(x_k) \tag{1}$$

where $\text{Emb}_s(\cdot)$ denotes the traditional textual embedding layer with positional embedding; the textual embedding representation $E_x \in$

$\mathbb{R}^{n \times d}$, and $d$ denotes the dimension of the textual embedding vector. [1]

*3.1.2 Visual Embedding.* We then leverage the pretrained Resnet-101 model [19] to extract initial visual representation, as shown as follows:

$$E_k^v = \text{Emb}_v(v_k) \tag{2}$$

where $\text{Emb}_v(\cdot)$ denotes the visual embedding layer with the pretrained Resnet-101 model followed by a single-layer Multi-layer Perception; the visual embedding representation $E_k^v \in \mathbb{R}^{7 \times 7 \times d}$.

### 3.2 Domain-shadow Aggregation with Modality-mixing Selection-voting Strategy

Incorporating the fine-grained and dispersed visual domain details into text has been proven effective in enhancing machine translation performance for DMNMT. To achieve this, we first attempt to aggregate domain-specific multi-modal information from the visual-to-textual direction to generate domain-shadow information through modality-mixing selection-voting strategy. Specifically, we first employ a fine-grained domain voting strategy to select domain closely-related visual details and domain loosely-related textual tokens in visual and textual modality spaces, respectively. Then we exchange the candidate fine-grained information across modality boundaries from visual to textual to generate modality-mixing domain representations.

*3.2.1 Modality-specific Encoders.* We first employ two modality-specific $L$-stacked Transformer layers to extract the textual and visual representations, respectively. The textual representation can be encoded as follows,

$$C_x^l = \text{TransEnc}_s^l(C_x^{l-1}) \tag{3}$$

where $\text{TransEnc}_s^l(\cdot)$ denotes the $l$-stacked Transformer layers, each Transformer layer consists of the multi-head attention (MHA) and feed forward networks (FFN); the layer index $l = 1, \ldots, L$, and when $l = 1$, we set $C_x^0 = E_x$; the textual representation $C_x^l \in \mathbb{R}^{n \times d}$.

Similarly, the visual representation is extracted as follows,

$$C_v^l = \text{TransEnc}_v^l(C_v^{l-1}) \tag{4}$$

where $\text{TransEnc}_v^l(\cdot)$ denotes the $l$-stacked Transformer layers, each Transformer layer consists of MHA and FFN; the layer index $l = 1, \ldots, L$, and when $l = 1$, we set $C_v^0 = E_v$; the visual representation $C_v^l \in \mathbb{R}^{49 \times d}$.

*3.2.2 Intra-modality Feature Selection.* To better capture domain-specific information from image-text data pair, we employ a domain feature selection mechanism within each modality space to adaptively select domain-related visual details and domain loosely-related textual details through intra-modality probability sampling. We design two domain-aware selective gating mechanisms in each stacked Transformer layer, as depicted as follows:

$$G_x^l = \sigma(W_x^l C_x^l + b_x^l) \tag{5}$$

$$G_v^l = \sigma(W_v^l C_v^l + b_v^l) \tag{6}$$

where $\sigma(\cdot)$ denotes sigmoid operation; $W_x^l, W_v^l, b_x^l$ and $b_v^l$ are trainable parameters; $G_x^l$ and $G_v^l$ represent the domain-aware textual

---

[1]For simplicity, the subscript k will be omitted in subsequent sections of this paper.

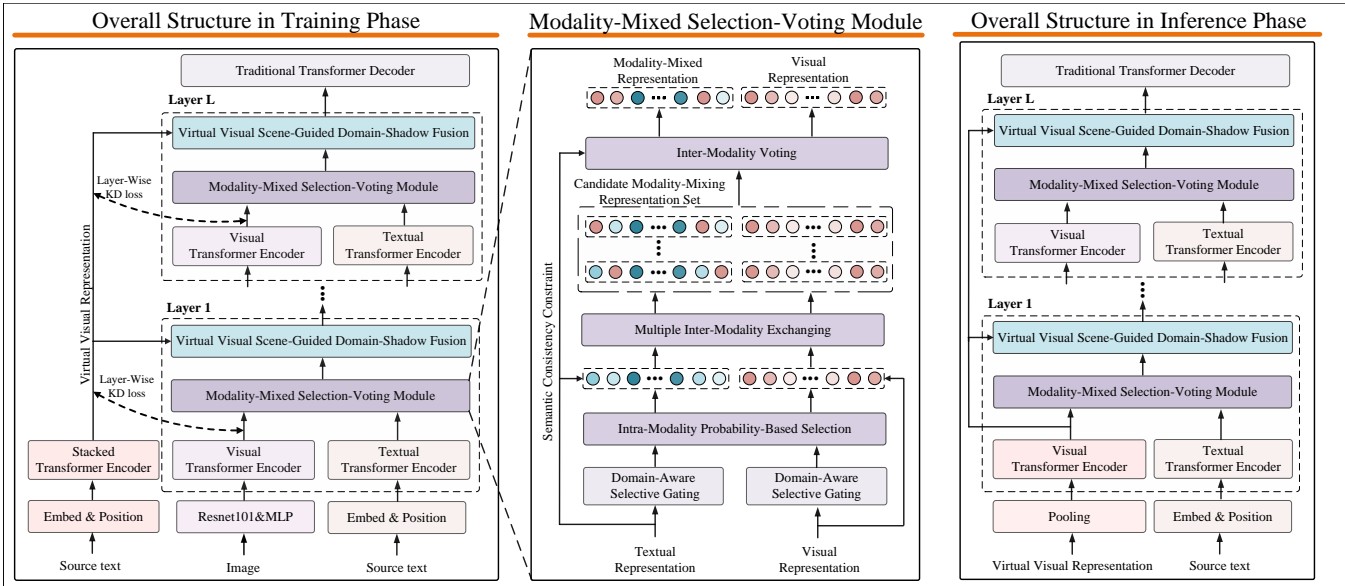

**Figure 2: The architecture of our model in the training and inference phases.**

and visual selected gating in the $l$-th Transformer layer, and $G_x^l$, $G_v^l \in \mathbb{R}^{n \times d}, \mathbb{R}^{49 \times d}$.

Then we employ a probability-based fine-grained $t$-sampling mechanism [2] to select domain-related visual details and domain loosely-related textual details dynamically, as shown as follows:

$$S_x^l = \text{MulSam}\left(\frac{\overline{G}_x^l}{\sum_{i=1}^{n}(\overline{G}_x^i)}, t\right) \tag{7}$$

$$S_v^l = \text{MulSam}\left(\frac{G_v^l}{\sum_{j=1}^{49}(G_v^j)}, t\right) \tag{8}$$

Where $\text{MulSam}(\cdot)$ denotes a probability-based fine-grained feature sample mechanism and $t$ is the number of sampling iterations; $\overline{G}_x^l$ represents the inverse sample probability, and $\overline{G}_x^l = 1 - G_x^l$. $S_x^l$, $S_v^l \in \mathbb{R}^{t \times d}, \mathbb{R}^{t \times d}$ denote the selected representations in the $l$-th Transformer layer.

*3.2.3 Inter-modality Exchanging and Voting.* We randomly exchange the selected representations across modality boundaries $\tau$ times to generate $\tau$ candidate modality-mixing domain representations by incorporating visual details into text. Then we select the smoothest exchange sample from all $\tau$ candidate exchange sets to obtain the modality-mixing representations.

**Inter-modality Exchanging.** We adopt a point-to-point exchange mechanism from visual to textual direction to exchange domain-related visual representations and domain loosely-related textual representations $\tau$ times to generate the modality-mixing domain representation set $\phi_x^l$, as depicted as follows,

$$\tilde{C}_x^l(r) = \text{Exch}\left(< S_{v,\alpha}^l \rightarrow S_{x,\beta}^l > | C_x^l, C_v^l, \tau\right) \tag{9}$$

<hr>

[2] We adopt $t$ times independent repeated sampling with replacement.

where $\text{Exch}(\cdot)$ denotes the point-to-point exchange operation; $< S_{v,\alpha}^l \rightarrow S_{x,\beta}^l >$ denotes an operation by replacing the $\alpha$-th vector of $S_v^l$ with the $\beta$-th vector of $S_x^l$. $\alpha, \beta = \text{randint}(1, t)$ represents a random sample from 1 to $t$; $\tilde{C}_x^l(r) \in \mathbb{R}^{n \times d}$ denotes a candidate exchange sample.

The candidate modality-mixing representation set can be obtained as follows,

$$\phi_x^l = \{\tilde{C}_x^l(r) | r = 1, \ldots, \tau\} \tag{10}$$

where $r$ is the index of candidate modality-mixing representation.

**Inter-modality Voting.** We only attempt to select a smoothest exchange sample $D_x^l$ from all $\tau$ candidate exchange set $\phi_x^l = \{\tilde{C}_x^l(r) | r = 1, \ldots, \tau\}$ to obtain the modality-mixing representation via the minimal change in KL divergence score, as described as follows,

$$D_x^l \triangleq \phi_x^l(u) \tag{11}$$

where $D_x^l \in \mathbb{R}^{n \times d}$ is the selected modality-mixing representation; $u$ is the selected index from 1 to $\tau$, and $u$ can be defined as follows,

$$u = \arg\min\left(\text{KL}\left(\tilde{C}_x^l(r) \middle\| C_x^l\right) | r = 1, \ldots, \tau\right) \tag{12}$$

where $\arg\min(\cdot)$ denotes the operation of calculating the minimum KL scores among all candidates; $\text{KL}\left(\tilde{C}_x^l(r) \middle\| C_x^l\right)$ denotes the KL divergence score between the $r$-th candidate modality-mixing representation $\tilde{C}_x^l(r)$ and the textual representation $C_x^l$ in the $l$-th layer.

This modality-mixing domain representation $D_x^l$ will be used to generate domain-specific multi-modal representation.

## 3.3 Virtual Visual Scene-guided Progressive Domain-shadow Fusion

To promote collaboration between modality-mixing domain representation and textual representation, we present a virtual visual

scene-guided domain-shadow fusion strategy in each Transformer layer.

### 3.3.1 Virtual Visual Scene Generation.
We employ a $P$-stacked Transformer encoder to generate virtual visual representation by taking the initial textual embedding as input, and we have that,

$$C_h = \text{StackTransEnc}_h(E_x) \tag{13}$$

where $\text{StackTransEnc}_h(\cdot)$ denotes the $P$-stacked Transformer encoder layers, each Transformer layer consists of MHA and FFN modules; $C_h \in \mathbb{R}^{n \times d}$ denotes the virtual visual scene representation generated from the source text.

### 3.3.2 Virtual Visual Scene-guided Domain Information Aggregation.
Then we adopt a virtual visual scene-guided domain information aggregation strategy to integrate modality-mixed domain representation and virtual visual representation through cross-modal gating fusion, as described as follows,

$$F_l = C_h + \delta_l \cdot D_x^l \tag{14}$$

where $F_l \in \mathbb{R}^{n \times d}$ denotes the fused multi-modal domain representation with the guidance of virtual visual representation at the $l$-layer; and the gating $\delta_l$ is calculated as follows:

$$\delta_l = \text{sigmoid}(W_h^l \cdot C_h + W_f^l \cdot D_x^l) \tag{15}$$

where $W_h^l \in \mathbb{R}^{d \times d}$ and $W_f^l \in \mathbb{R}^{d \times d}$ are learnable parameters, $\text{sigmoid}(\cdot)$ represents the element-wise sigmoid transformation.

### 3.3.3 The Progressive Domain-shadow Fusion Strategy.
To comprehensively enhance domain-shadow fusion, we adopt an adaptive cross-layer domain-shadow fusion strategy to progressively integrate modality-mixed domain-shadow information layer-by-layer with the guidance of virtual visual representation, as demonstrated in Figure 2.

## 3.4 Training Schedules in Training and Testing Stages

The virtual visual scene plays a critical role in domain-specific multi-modal representation. To further enhance domain-shadow fusion, we employ a multi-layer visual distillation mechanism aimed at capturing multi-grained domain visual details at each Transformer layer. During the training stage, the model's parameters are optimized using a joint loss function, which is detailed as follows:

$$loss = loss_{CE} + \sum_n^l \theta_n loss_h^n \tag{16}$$

where $loss_{CE}$ represents traditional machine translation loss and $\theta_n$ is the layer-wise loss hyper-parameter; the visual distillation loss $loss_h^n$ is defined as follows:

$$loss_h^n = \text{KL}(C_h || C_v^n) \tag{17}$$

where $\text{KL}(\cdot)$ denotes the operation of calculating KL scores; $(\cdot || \cdot)$ represents visual-centric distillation by randomly up-sampling 49 times in the length dimension to transform $C_h$ into the length dimension of $C_v^n$.

Furthermore, in the inference stage, our model can utilize virtual visual information instead of the ground truth visual representation as visual input to provide multi-grained visual representation for machine translation. Therefore, our model can adapt to text-only scenarios.

## 4 EXPERIMENTS

### 4.1 Experimental settings

**Datasets.** We conduct experiments on three benchmark MNMT datasets, including two domain-specific datasets, Fashion-MMT [33] and EMMT [46], and one general-domain dataset, Multi-30k [7]. Specifically, 1) Fashion-MMT is a MNMT dataset containing two sub-datasets: Fashion-MMT(clean) and Fashion-MMT(large) in the E-commerce domain. The Fashion-MMT(clean) dataset is composed of 40,000 image-text pairs. Each data pair includes an English description, one or more images, and a manually edited Chinese translation. The Fashion-MMT(large) dataset contains 114,257 image-text pairs, where each data pair contains one or more images, an English description, and a noisy Chinese sentence translated by a text-only SOTA model. 2) EMMT: The EMMT dataset is a real-world e-commercial dataset collected from TikTok Shop and Shopee. It comprises 22,500 annotated triplets, where each triplet consists of an English product description, a manually annotated Chinese translation, and a corresponding image. The test set is selected by professional annotators, comprising examples that are challenging to translate without corresponding images. 3) Multi-30k: Multi-30k is the widely used benchmark for MNMT tasks, covering a variety of general-domain scenarios. The Multi-30k dataset contains 29k text-image data pairs for training and 1014 data pairs for validation. And we follow standard evaluation setup to report the results on three test splits, Test2016, Test2017 and MSCOCO.

**Evaluation Metrics.** We utilize three types of metrics to evaluate the performance of machine translation, including BLEU [29], METEOR [5], and BLEURT. BLEURT is a robust noise evaluation metric proposed by [32], demonstrating a strong correlation with human evaluation. We also report the average scores and Student's t-test (T.TEST) scores by running each model three times.

**Implementation Details.** We employ byte pair encoding (BPE) segmentation with 8k, 10k, and 6k merge operations for the Fashion-MMT, EMMT and Multi-30k datasets, respectively. The vocabulary sizes are 8880-2936 tokens for the Fashion-MMT dataset, 10407-9799 tokens for the EMMT dataset, 5644-5876 tokens for the Multi-30k (En-De) translation task, and 5644-5972 tokens for the Multi-30k (En-Fr) translation task. We utilize the pre-trained CLIP model [32] to represent textual and visual features into a shared multi-modal space, thereby obtaining the most semantic-related image features corresponding to its text for Fashion-MMT dataset. Furthermore, our model consists of 4 stacked encoders and 4 stacked decoders based on the Transformer-based seq2seq framework for all datasets.

## 4.2 Comparison results on three MNMT datasets

### 4.2.1 Comparison Results on Domain-specific Fashion-MMT and EMMT Datasets.
We first carry out experiments on two domain-specific Fashion-MMT and EMMT datasets, the comparison results

**Table 1: Comparison results on domain-specific Fashion-MMT and EMMT datasets. The best scores are highlighted in bold. ↑ indicates that the improvement achieved by our model over the best result of our reproduced MNMT models is statistically significant, with a p-value < 0.01.**

| Model | En→Zh task | | | | | | | | |
| | Fashion-MMT(clean) | | | Fashion-MMT(large) | | | EMMT | | |
| | BLEU | METEOR | BLEURT | BLEU | METEOR | BLEURT | BLEU | METEOR | BLEURT |
|---|---|---|---|---|---|---|---|---|---|
| Existing DMNMT and MNMT Models | | | | | | | | | |
| Transformer[35] | 40.61 | 35.77 | - | 41.21 | 35.91 | - | 39.07 | - | 54.24 |
| Multimodal Graph[42] | 40.70 | 35.45 | - | 41.49 | 35.95 | - | - | - | - |
| UPOC(MTLM+ISM)[33] | 41.38 | 35.68 | - | 43.00 | 36.68 | - | 40.60 | - | 48.55 |
| 2/3-Triplet[46] | 41.38 | - | - | 42.33 | - | - | 42.03 | - | 57.60 |
| UVR-NMT[9] | - | - | - | - | - | - | 37.82 | - | 52.99 |
| Our Reproduced MNMT Models | | | | | | | | | |
| Doubly-ATT[2] | 40.46 | 35.78 | 58.94 | 42.97 | 36.87 | 60.92 | 41.69 | 33.17 | 55.67 |
| Gated Fusion[39] | 39.97 | 35.27 | 58.78 | 41.78 | 36.24 | 60.97 | 40.97 | 32.98 | 54.97 |
| Selective attention[22] | 40.52 | 35.67 | 58.87 | 42.76 | 37.08 | 61.14 | 41.54 | 33.25 | 56.23 |
| Our Ground-truth and Virtual Visual Model | | | | | | | | | |
| Our model(G) | **41.57**↑ | 36.25 | 60.47 | **44.43**↑ | **38.32**↑ | 62.03 | **43.91**↑ | **34.87**↑ | **58.71**↑ |
| Our model(H) | 41.52 | **36.34**↑ | **60.53**↑ | 44.40 | 38.30 | **62.13**↑ | 43.84 | 34.77 | 58.68 |

**Table 2: Comparison results on the En→De and En→Fr translation tasks on the Multi30k dataset. The best scores are highlighted in bold. ↑ marks that the improvement achieved by our model over the best result of our reproduced MNMT models is statistically significant, with a p-value < 0.01. MultiAtt, GatFus, and SelAtt denote Multimodal Self-attention, Gated Fusion, and Selective Attention, respectively. The MET refers to the METEOR evaluation metric.**

| Models | Multi30k En→De | | | | | | Multi30k En→Fr | | | | | |
| | Test2016 | | Test2017 | | MSCOCO | | Test2016 | | Test2017 | | MSCOCO | |
| | BLEU | MET | BLEU | MET | BLEU | MET | BLEU | MET | BLEU | MET | BLEU | MET |
|---|---|---|---|---|---|---|---|---|---|---|---|---|
| Existing MNMT Models | | | | | | | | | | | | |
| RMMT[39] | 41.45 | - | 32.94 | - | 30.0 | - | 62.12 | - | 54.39 | - | 44.52 | - |
| IKD-MMT[30] | 41.2 | 58.9 | 33.8 | 53.2 | 30.1 | 48.9 | 62.5 | 77.2 | 54.8 | 71.8 | - | - |
| VALHALLA(M)[25] | 42.6 | - | 35.1 | - | 30.7 | - | 63.1 | - | 56.0 | - | 46.4 | - |
| MDA[15] | 42.0 | 59.4 | 34.1 | 52.5 | 30.4 | 49.6 | 62.4 | 77.2 | 54.1 | 72.1 | 46.5 | 66.7 |
| EDC[34] | 42.0 | 60.2 | 33.4 | 53.7 | 30.0 | 49.6 | 62.9 | 77.2 | 55.8 | 72.0 | 45.1 | 64.9 |
| Our Reproduced NMT and MNMT Models | | | | | | | | | | | | |
| Transformer[35] | 40.78 | 59.45 | 32.76 | 51.37 | 28.76 | 48.22 | 60.48 | 75.83 | 53.12 | 70.85 | 43.75 | 64.48 |
| MultiAtt[40] | 41.51 | 58.78 | 32.96 | 51.98 | 29.43 | 48.42 | 60.96 | 74.98 | 54.17 | 71.22 | 44.35 | 64.65 |
| GatFus [39] | 41.55 | 58.64 | 32.87 | 51.87 | 29.59 | 48.71 | 61.46 | 75.27 | 53.93 | 71.34 | 44.21 | 64.26 |
| SelAtt[22] | 42.03 | 59.07 | 34.05 | 52.78 | 30.27 | 49.34 | 61.78 | 76.23 | 54.27 | 72.25 | 44.89 | 65.22 |
| Our Ground-truth and Virtual Visual Model | | | | | | | | | | | | |
| Our Model(G) | 42.83 | **60.51**↑ | 35.20 | **54.51**↑ | **31.21**↑ | **51.27**↑ | 63.24 | **77.84**↑ | 56.29↑ | **73.22**↑ | 46.83↑ | 67.40 |
| Our Model(H) | **42.85**↑ | 60.48 | **35.31**↑ | 54.48 | 31.17 | 51.25 | **63.27**↑ | 77.75 | 56.19 | 73.20 | 46.71 | **67.42**↑ |

are presented in Table 1. "Our model(G)" and "Our model(H)" denote our proposed model that utilizes ground-truth images and virtual visual information, respectively. We can see that 1) Our model achieves a significant improvement over all the other SOTA DM-NMT approaches across three types of evaluation metrics on two datasets. 2) The proposed approach significantly outperforms text-only transformer approach, confirming the effectiveness of visual information for machine translation. 3) Both Our model(G) and Our model(H) achieve comparable machine translation scores on all test sets. It demonstrates the effectiveness of our proposed approach for DMNMT.

*4.2.2 Comparison Results on Multi30k Dataset in General Domain.* To further confirm the robustness of our proposed method, we conduct additional experiments on the Multi30k dataset. The results of English-to-German and English-to-French translation tasks are presented in Table 2. The findings are as follows: 1) In comparison to existing MNMT models, our method achieves SOTA scores in the BLEU and METEOR metrics on the test2016, test2017, and MSCOCO test sets. 2) Compared to our reproduced NMT and MNMT models, our approach demonstrates significant improvements under the same parameter and environment settings. Furthermore, we also conduct significance tests between our reproduced models

**Table 3: Ablation study on different modules of our proposed model.**

| Fashion-MMT En→Zh task | | | | | | | |
|---|---|---|---|---|---|---|---|
| Different modules of our model | | | Our model | Fashion-MMT(clean) | | Fashion-MMT(large) | |
| Cross-modal exchanging module | Cross-modal fusion module | Virtual visual scene generation module | | BLEU | BLEURT | BLEU | BLEURT |
| - | √ | √ | Our model(G) | 39.59 | 57.27 | 41.97 | 59.54 |
| | | | Our model(H) | 39.63 | 57.59 | 41.88 | 59.47 |
| √ | - | √ | Our model(G) | 41.29 | 59.88 | 44.09 | 61.47 |
| | | | Our model(H) | 41.17 | 59.72 | 44.13 | 61.60 |
| √ | √ | - | Our model(G) | 40.78 | 58.69 | 43.17 | 60.25 |
| √ | √ | √ | Our model(G) | 41.57 | 60.47 | 44.43 | 62.03 |
| | | | Our model(H) | 41.52 | 60.53 | 44.40 | 62.13 |

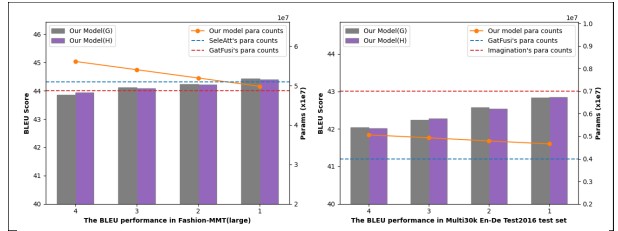

**Figure 3: The BLEU and parameters of our proposed virtual visual scene generation module on different layers. The horizontal axis represents the number of layers in the virtual visual scene generation module. SeleAtt, GatFusi and Imagination denote the Selective attention [22], Gated fusion [42] and Imagination [8] models, respectively.**

and our approach. The significance test results indicate that our model achieves a statistically significant improvement over these models (p-value < 0.01). 3) Our model(G) and Our model(H) also achieve comparable results on these three test sets. This confirms the effectiveness of our virtual scene generation module. Furthermore, it is noteworthy that the MSCOCO test split, which includes sentences with ambiguous verbs and out-of-domain samples from the COCO Captions dataset, is often challenging for MNMT models. However, our model performs exceptionally well on this test set, which suggests it can effectively employ visual information to handle ambiguity through our proposed modality exchanging mechanism.

## 4.3 Ablation Study

*4.3.1 Ablation Study on Different Modules.* We first conduct ablation studies on different modules to demonstrate the effectiveness of the modules in our proposed approach. The experiment results are shown in Table 3. The conclusions could be drawn as follows: 1) Replacing the cross-modal exchanging module with original textual representation causes a significant performance drop on Fashion-MMT(small) and Fashion-MMT(large) datasets. 2) Replacing the cross-modal fusion module with a pooling-addition operation also causes a remarkable machine translation performance decline on two tasks. 3) Removing virtual visual scene generation module causes huger BLEU and BLEURT scores drop than replacing cross-modal fusion module, which confirms that the visual scene

generation module plays a significant role in guiding the process of multimodal fusion. The results confirm the validity of the proposed modules in domain-specific translation tasks, especially for cross-modal exchanging module.

*4.3.2 Impact of Parameter Counts on the Virtual Visual Scene Generation Module.* We then conduct experimental analysis on the impact of parameter counts on the virtual visual scene generation module to investigate its model compression capability, as shown in Figure 3. Where the rectangular boxes represent the BLEU scores, the dashed and dotted lines represent the model's parameter counts. We evaluate the model compression capability for several types of modules with comparable parameter counts, including our proposed *P*-stacked Transformer encoder, the traditional selective attention [22], the Gated Fusion [39], and Imagination [8] modules. It can be included as follows: 1) Our proposed virtual visual scene generation module still exhibits strong machine translation performance even when the number of stacked Transformer layers is set to 1. 2) To further demonstrate the robustness and generalization of our virtual visual generation module, we also conduct experiments in the widely-used Multi30k(En-De) Test2016 task. The experimental results showcase that our *P*-stacked Transformer encoder achieves the highest BLEU scores when the layer number is 1, with the fewest number of parameters. These findings confirm the effectiveness of model compression, highlighting its potential for virtual visual representation.

*4.3.3 Validity of Image Information in the Inference Phase.* To investigate the robustness of our model for visual information, we further examine the validity of images in machine translation by adversarial evaluation [6], as demonstrated in Table 4. Specifically, we replace ground-truth images by the following three types of images, including blank image (BlkImg), randomly selected image (RSImg), noisy image (NsImg). Furthermore, we evaluate machine translation performance with visual adversarial evaluation by considering the following three additional scenarios: 1) Text-BlkImg data pair scenario; 2) Text-RSImg data pair scenario, and 3) Text-NsImg data pair scenario. The experimental results demonstrate that our approach exhibits significant robustness across several visual noisy scenarios. Furthermore, our model(H) achieves slightly higher BLEU and METEOR scores in all three noisy multi-modal scenarios, as it does not rely on noisy visual images. This suggests

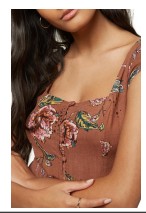
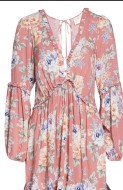

SRC:  intricately **sketched blooms** and dainty **covered buttons** make this **cap sleeve top** so romantic.
REF:  精致的**手绘花朵**和优雅的**包覆纽扣**使这款**沉肩袖上衣**尽显浪漫。
Transformer:复杂细致的**花朵图案**和精致的**包覆纽扣**使得这件**短袖上衣**充满了浪漫气息。
Gated Fusion:错综复杂的**草绘花朵**和精致的**带盖纽扣**使这款**帽袖上衣**非常浪漫。
Selective Attention:精致的**草绘花朵**和精致的**包覆纽扣**使这款**帽袖上衣**显得非常浪漫。
GPT4:复杂精细的**花卉草图**和小巧的**包布纽扣**使这款**短袖上衣**显得如此浪漫。
Our model(G):精致的**手绘花朵**和精致的**包覆纽扣**使这款**沉肩袖上衣**如此浪漫。
Our model(H):复杂精致的**手绘花朵**和精致的**包覆纽扣**使这款**落肩袖上衣**如此浪漫。

SRC: a vintage inspired rose print on a **dusty pink ground** sets the stage for this romantic frock featuring **delicate frills** and covered buttons .
REF: **飘逸的粉色布料**上复古的玫瑰印花为这件浪漫连衣裙奠定了基础，这件连衣裙上饰有**精致的褶边**和包覆纽扣。
Transformer: 一幅复古风格的玫瑰图案印在**浅粉色的地面**上，为这件浪漫连衣裙铺设了舞台，上面装饰有**精致的荷叶边**和盖住的纽扣。
Gated Fusion: 这款浪漫的连衣裙以**精致的荷叶边**和带盖纽扣为特色，在**尘土飞扬的粉色地面**上设计了复古灵感的玫瑰印花。
Selective Attention: 这款浪漫的连衣裙采用**精致的荷叶边**和带盖纽扣的设计，复古风格的玫瑰印花在**尘土飞扬的粉色地面**上占据了基础。
GPT4: 一种复古风格的玫瑰图案印在**浅粉色的底色**上，为这件充满浪漫气息的连衣裙设定了基调，连衣裙特点包括**精致的荷叶边**和包布纽扣。
Our Model(G): 复古风格的玫瑰印花在**飘逸的粉色布料**上，为这款浪漫的连衣裙奠定了基础，连衣裙饰有**精致的褶边**和包覆纽扣。
Our Model(H): 一款复古风格的玫瑰印花在**飘逸的粉色布料**上，为这件浪漫的连衣裙设定了基础，连衣裙饰有**精致的褶边**和包覆纽扣。

**Figure 4: Two examples of domain-specific machine translation. "SRC" and "REF" denote the source and reference sentence.**

that noisy images indeed have a negative impact on machine translation.

**Table 4: Different multimodal scenarios in testing process.**

| Multi-modal data | Fashion-MMT En→Zh | | | |
| | Fashion-MMT (clean) | | Fashion-MMT (large) | |
| | BLEU | BLEURT | BLEU | BLEURT |
|---|---|---|---|---|
| Text-BlkImg scenario | 41.32 | 60.25 | 44.11 | 61.89 |
| Text-RSImg scenario | 41.41 | 60.18 | 44.25 | 61.73 |
| Text-NsImg scenario | 41.44 | 60.39 | 44.32 | 61.97 |
| **Text-only scenario (Our model(H))** | 41.52 | 60.53 | 44.40 | 62.13 |
| **Our model(G)** | 41.57 | 60.47 | 44.43 | 62.03 |

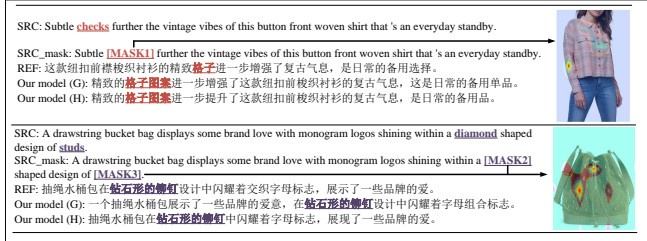

SRC: Subtle **checks** further the vintage vibes of this button front woven shirt that 's an everyday standby.
SRC_mask: Subtle **[MASK1]** further the vintage vibes of this button front woven shirt that 's an everyday standby.
REF: 这款纽扣前襟梭织衬衫的精致**格子**进一步增强了复古气息，是日常的备用选择。
Our model (G): 精致的**格子图案**进一步增强了这款纽扣前襟织衬衫的复古气息，这是日常的备用单品。
Our model (H): 精致的**格子图案**进一步提升了这款纽扣前襟织衬衫的复古气息，是日常的备用品。

SRC: A drawstring bucket bag displays some brand love with monogram logos shining within a **diamond** shaped design of **studs**.
SRC_mask: A drawstring bucket bag displays some brand love with monogram logos shining within a **[MASK2]** shaped design of **[MASK3]**.
REF: 抽绳水桶包在**钻石形的铆钉**设计中闪耀着交织字母标志，展示了一些品牌的爱。
Our model (G): 一个抽绳水桶包展示了一些品牌的爱意，在**钻石形的铆钉**设计中闪耀着字母组合标志。
Our model (H): 抽绳水桶包在**钻石形的铆钉**中闪耀着字母标志，展现了一些品牌的爱。

**Figure 5: Two examples of domain-related textual information being masked during the inference phase. SRC_mask denotes the masked source sentence.**

## 4.4  Case Study

Figure 4 depicts the translation of two domain-specific cases of Fashion-MMT dataset. Colors highlight improvement. In these examples, our proposed approach can translate domain-specific terms correctly, such as "sketched blooms," "cap sleeve top," "dusty pink ground," etc. The text-only model fails to generate the domain term "cap sleeve top" without the aid of visual information. This confirms the importance of visual information for DMNMT. Moreover,

our model(H) still can translate domain-specific terms correctly, such as " 手绘花朵,"" 飘逸的粉色布料," and " 精致的褶边." It demonstrates the effectiveness of virtual visual scene generation for domain-specific term translation.

## 4.5  Visual Analysis

To explore the robustness of our proposed model and its ability to use image information for improving domain-specific translations when key text details are missing, we replaced certain domain terms with [MASK] during testing. As illustrated in Figure 5, by visualizing the attention weights of the Vision Transformer, we masked words like "checks", "diamond" and "studs". These words, which vary greatly between specific and general domains, pose a translation challenge without corresponding image cues. In Case 1, despite masking the word "checks," which differs significantly from its usual expression, our model achieved accurate translation by focusing on related image areas. Intriguingly, in Case 2, accurately translating masked domain-related words "diamond" and "studs" is significantly challenging without corresponding images. The focused image information by our model indicates it can precisely identify relevant areas for these domain-specific terms, enhancing translation performance.

## 5  CONCLUSION

This paper has addressed the multi-focus challenges associated with visual-to-textual interaction in DMNMT, introducing a virtual visual scene-guided progressive domain-shadow fusion approach to enhance the model's capability in perceiving fine-grained, domain-specific visual details. Extensive experiments conducted on three benchmark datasets demonstrate that our proposed approach surpasses SOTA MNMT models, achieving significant improvements across all machine translation tasks. Moreover, our in-depth ablation studies and adversarial evaluation underscore the robustness and generalizability of our approach across both general and domain-specific contexts, even under noisy or text-only conditions. Additionally, the virtual visual scene generation module has shown a remarkable potential for model compression, indicating its viability for real-world applications.

# ACKNOWLEDGMENTS

This work is supported by National Natural Science Foundation of China(No. 62366025), Natural Science Foundation project of Yunnan Science and Technology Department (No. 202301AT070444), and Yunnan provincial major science and technology special plan projects (No. 202202AE090008-3).

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
