# OpenReview forum: "Virtual Visual-Guided Domain-Shadow Fusion via Modal Exchanging for Domain-Specific Multi-Modal Neural Machine Translation"
_acmmm.org/ACMMM/2024/Conference — MM2024 Poster_

### Official Review · Reviewer_a55b · 2024-05-24

**Rating:** 4
**Confidence:** 3

**Summary:**

In order to realize domain-specific multi-modal neural machine translation, this paper proposes a novel modality-mixing selection-voting strategy to integrate visual details into text representations through intra-modality selection and inter-modality exchanging. After that, a virtual visual scene-guided progressive domain-shadow fusion is developed to generate domain-specific multi-model representation, so that the machine translation performance can be improved.

**Strengths:**

1.  The paper is well written and easy to follow. The figures are clear and easy to understand.
2.  Utilizing modality-mixing selection-voting strategy to incorporate visual details into text is very interesting. And the related operations are technical reasonable.
3.  Extensive experiments and ablation studies are conducted, and the results demonstrate the effectiveness of the proposed method.

**Limitations:**

Just two small questions:

1.	I can understand “visual-to-textual interaction in a specific domain often exhibits multi-focus characteristics”. But I am little confused that why dose this bring challenges or negative effects to the multi-modal machine translation? Could the authors explain it in more details?

2.	In my own opinion, the Virtual Visual Scene-guided Progressive Domain-shadow Fusion module is just a gated-fusion process, could the authors explain its underlying mechanism for tackling the multi-focus challenges (as claimed in 249th-250th rows)?

Overall, I vote for borderline accept for this paper.

**Suitability:**

3

---

### Official Review · Reviewer_1yMQ · 2024-05-27

**Rating:** 3
**Confidence:** 2

**Summary:**

This paper focuses on domain-specific multi-modal neural machine translation, which faces the challenges of visual-to-textual interaction in a specific domain, frequently exhibiting multi-focus characteristics. To address the challenge, this paper proposes a  a virtual visual scene-guided progressive domain-shadow fusion approach, which includes domain-shadow aggregation with modality-mixing selection-voting strategy and  virtual visual scene-guided progressive domain-shadow fusion. Extensive experimental results on public datasets demonstrate the effectiveness of proposed method in the face of missing modal data.

**Strengths:**

1.The method significantly enhances the model’s capability in perceiving fine-grained, domain-specific visual details, addressing the multi-focus challenges of visual-to-textual interaction in DMNMT.

2.Comprehensive experimental analysis has shown that this method exhibits significant robustness and generalizability across various scenarios.

**Limitations:**

1.The expression in the paper is too casual, and the structure is not reasonable. For instance, Lines 129 and 307 even extend beyond the line, causing disorder. Incorrect use of double quotes and single quotes in lines 895 and 901. “... presented in Tables 2” should be “... presented in Table 2”.

2.The proposed method lacks novelty. Several modules proposed by the authors are very common. Where is your advantage?

3.In line 447, why are textual embeddings used as input to generate virtual visual representations?

4.In line 455, the modality-mixed domain representation already includes visual and textual features. Why is it necessary to combine it with the virtual visual representation obtained in the previous step?

5.The methods [1,2] are not included in the comparative experiments. Provide reasons for their exclusion.

[1]Progressive modality-complement aggregative multitransformer for domain multi-modal neural machine translation
[2]Dual transfer learning for neural machine translation with marginal distribution regularization

**Suitability:**

3

---

### Official Review · Reviewer_7WuD · 2024-06-05

**Rating:** 5
**Confidence:** 2

**Summary:**

This paper is about improving domain-specific translation by exploiting multimodal information, i.e., images associated to text.
A complex transformer-based model is proposed to fuse visual and textual representations that enable focusing on domain-specific aspects of the multimodal content and the level of interaction between the different modalities.

**Strengths:**

The proposed model is an original and sound approach to the problem. Its ideas can find application in other tasks, e.g., multi-modal multi-language classification.
The technical parts of the paper are ok (see limitations for a more general comment on the paper content).
Experimental setup is clearly presented and solid.
Multiple dataset are used to test the methods.
To the best of my knowledge the methods used as baselines are a sound choice for the task.
Results show a statistically significant improvement, supporting the claims and the proposed methods.
Ablation study is interesting.
Code to replicate the experiments is available.

**Limitations:**

This is a technical paper, with a "cold" way to present its content. This could limit its reach in the MM community.
Title is almost sequence of keywords and it is complex, somewhat cryptic to the non-expert, and does not clearly conveys the task and contribution of the paper.
A title should be simpler to read, and convey a general idea of the work, that is then clearly presented in the abstract and in the paper with increasing detail and precision.
The whole paper is a bit hard to read, as it is very repetitive in the use of multiword terms that are not properly explained until the very central part of the paper, leaving the reader with partial information for the most part of the reading. For example the expression "virtual visual scene-guided domain-shadow multi-modal fusion" appears more than 10 times, and it can be understood only from the technical section of the paper. Using a bit of the first part of the paper to break it in its various parts and giving an intuitive explanation of the concepts involved would help a lot the readability.
A computational cost comparison is missing.

**Suitability:**

3

---

### Meta-Review · Area_Chair_BedF · 2024-07-02

**Recommendation:** Accept (Poster)
**Confidence:** 3

**Metareview:**

Two reviewers are in favor of accepting (weak) the paper while one tends towards rejecting (borderline).
I suggest accepting this paper because the authors adequately addressed most of the reviewers' concerns in their rebuttal.
Regarding the reviewer reporting a final rating of borderline reject, his final rating justification says that the "author partially addressed my concern" while keeping his original score.